

# Determination of the time scale of photoemission from the measurement of spin polarization

**Mauro Fanciulli**[1,2][*][†] **and J. Hugo Dil**[1,2]

**1** Institut de Physique, École Polytechnique Fédérale de Lausanne,
1015 Lausanne, Switzerland
**2** Photon Science Division, Paul Scherrer Institut, 5232 Villigen, Switzerland

[*] mauro.fanciulli@u-cergy.fr
[†] Current affiliation: Laboratoire de Physique des Matériaux et des Surfaces,
Université de Cergy-Pontoise, 95031 Cergy-Pontoise, France

## Abstract

The Eisenbud-Wigner-Smith (EWS) time delay of photoemission depends on the phase term of the matrix element describing the transition. Because of an interference process between partial channels, the photoelectrons acquire a spin polarization which is also related to the phase term. The analytical model for estimating the time delay by measuring the spin polarization is reviewed in this manuscript. In particular, the distinction between scattering EWS and interfering EWS time delay will be introduced, providing an insight in the chronoscopy of photoemission. The method is applied to the recent experimental data for Cu(111) presented in *M. Fanciulli et al., PRL 118, 067402 (2017)*, allowing to give better upper and lower bounds and estimates for the EWS time delays.

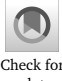

# 1 Introduction

The description of time in quantum mechanics poses several fundamental difficulties, ultimately because the operators that can be associated to the variable of time are Hermitian, but not self-adjoint [1]. As a consequence, the corresponding eigenvalues are not necessarily real quantities, which is a requirement for physical observables. Therefore time is often considered just as a classical parameter of the quantum mechanics equations. In the experiments, one usually takes into account the time duration only when looking at "macroscopic" classical time scales, in the sense of dynamical behaviour of complex collective phenomena, as in a pump and probe experiment. The flow of time can then be seen as in classical daily experience. On the other hand, when considering very basic quantum processes, time is just not explicitly taken into account, and changes of states of a system are considered to be *instantaneous*. Indeed, down to the picosecond ($10^{-12}$ s) or even femtosecond ($10^{-15}$ s) regime, processes such as tunneling, radiation-matter interaction, or the wavefunction collapse itself, can be safely considered to be instantaneous. This pragmatic solution has been the workaround of the issue for many decades, since it was anyway impossible to try to measure such extremely short time durations.

However, apart from the fact that the instantaneousness of a process is not satisfactory, advances in laser technology in the past fifteen years have opened up the possibility to investigate electron dynamics in matter at a fundamental level, in a new field called *attosecond physics* [2–6]. Attoseconds (as, $10^{-18}$ s) are the natural time scale of fundamental atomic processes. Very naively, in 1 as light travels for about 3 Å, which is of the order of an atomic size. Also, it is instructive to consider the unit of time in atomic units $t_{a.u.} = \frac{\hbar}{\alpha^2 m_e c^2} \approx 24$ as, which corresponds to the time it takes an electron to perform one orbit of the Hydrogen atom in the Bohr model.

The processes that are tackled in attosecond physics involve radiation-matter interaction, and can be divided mainly into two families: radiation-driven tunneling ionization, and photon absorption. In the first case, the interaction with the light electric field bends the atomic potential so that electronic bound states are allowed to tunnel outside the atom. This allows to study the equivalent travel time of an electron during the tunneling process, a fundamental question in the history of quantum physics [7] that has not yet a fully accepted answer [8–13]. In the second case, the photon provides enough energy to an electron to escape from the bound state, a process that is usually called *photoionization* in the case of atomic systems, and *photoemission* in the case of condensed matter targets, which is the process considered in this publication.

All the current attosecond-resolved photoemission techniques are based on pump-probe setups that rely on a $\approx 100$ *as* ultraviolet (UV) laser pump and a phase-controlled few-cycle femtosecond infrared (IR) laser probe, which are temporally closely correlated because of the high harmonic generation process they rely on [14–16]. There exist two main different techniques: attosecond streaking [14, 16–19], and "reconstruction of attosecond beating by interference of two-photon transitions'" (RABBITT) [15, 20–22]. Several variation of these techniques are possible, with different proposed algorithms for the data analysis. In particular, it is crucial to be able to disentangle the time information of the probed material from additional time delays introduced by the specific measurements, which are of different kind, and not always easy to identify. Here it is only important to underline the following: because of their pump-probe nature, all the attosecond-resolved spectroscopy experiments can only probe a *relative* time delay between some electronic state under consideration and some other *experimental reference*. Such reference could be a different electronic state of the system [18], or a different system [23], or the same state under a different experimental condition [24], but no information can be directly extracted about the *absolute* time delay, because of the

nature of a pump-probe experiment. Only recently, a way to access absolute time delays has been proposed, which however has to rely on theoretical result as an input [25,26]. The word "absolute" should not generate confusion: a time delay $\tau$ describes a time duration, i.e. the interval in time with respect to time-zero. In particular, time-zero will be the instant when the considered process begins, and is not necessarily easy to define in the experiment [19]. Still, the point is that attosecond-resolved experiments can directly measure only time delay *differences* $\Delta\tau$ between different processes. The concept of time delay that is used to describe the time scale of the photoemission process will be summarized in Section 1.1. In particular it will be shown how it depends on the *phase term* of the matrix element associated to the photoemission transition under consideration.

In conventional photoemission spectroscopy one measures the energy of the emitted electrons from a material after the interaction with electromagnetic radiation. An energy-resolved measurement exploits the Fermi golden rule, for which the photoemission intensity is given by $I \propto |M_{fi}|^2$. $M_{fi}$ is the matrix element of the interaction Hamiltonian $M_{fi} = \langle \psi_f | \hat{H}_{int} | \psi_i \rangle$, with $|\psi_i\rangle$ the initial Bloch state and $|\psi_f\rangle$ a time-reversed LEED state in the one-step description of photoemission. The matrix element is a complex quantity $M_{fi} = Re^{i\phi}$, but because of the modulus square in the Fermi golden rule the information about the phase term is lost. From this, it looks like the only way to access phase shifts and therefore time delays would be to use pump and probe setups, where one does not look at the eigenstates of a system, but at all its possible time-dependent responses. However, if other quantum numbers do depend on the phase of the matrix elements, it will still be possible to access the phase information and thus the time information by measuring the corresponding physical observables without explicit time resolution. Here it will be argued that this is indeed the case.

An example is the *momentum* of the photoelectrons: it can be shown [27] that the differential photoemission cross-section, i.e. the angular distribution of the electrons, does actually depend on the phase term. Therefore one can, in principle, extract the phase information by measuring the angular distribution of the emitted electrons. This is a complex experiment, but can be performed for atomic and molecular levels [28–30]. UV photoelectron diffraction (UPD) experiments show that it is possible to retrieve the phase information by circular dichroism in orbitals from non-chiral molecules [31,32]. However it is intrinsically very difficult for dispersive states of a solid, since in this case the angular distribution unavoidably also depends on the energy-momentum dispersion relationship. In fact, in angle-resolved photoemission spectroscopy (ARPES) experiments one measures the angular and energetic distribution of the photoelectrons in order to reconstruct the band structure of the material, but any modulation and asymmetry of intensity where the phase term plays a role are often simply disregarded as "matrix element" effects.

Another quantum number that carries the phase information is the *spin* of the photoelectrons. Indeed, also the spin polarization of a beam of electrons emitted at a certain angle is a function of the phase of the matrix elements [27,33]. Therefore it is in principle possible to retrieve the phase information from the spin polarization in photoemission from spin-degenerate states. The dependence of the spin polarization on the phase term will be summarized in Section 1.2.

The determination of the phase information from the measurement of spin polarization has been extensively performed in atomic photoionization (see Refs. [34,35] and references therein). To some extent also the case of photoemission from solids has been considered [36, 37], but the lack of energy and momentum resolution has been a limitation in the past. With the development of setups with better resolution, then, the focus has been put mainly on the study of materials where the spin polarization is an interesting physical property of the initial state. However, the possibility to extract information on time delays from the determination of phases has been recently proposed in the literature [38]. The estimate of EWS time delays

in photoemission without a direct time-resolution in the experiment from the measurement of the spin polarization of electrons emitted from spin-degenerate dispersive states of a solid has been performed only very recently [39, 40]. However, the analytical relationship between spin polarization and time delay introduced in Ref. [39] lacked detail and, more importantly, no clear distinction between scattering and interfering time delays was made. Here we expand this model and discuss the physical nature of the time delays.

## 1.1 The Eisenbud-Wigner-Smith (EWS) time delay in photoemission

The concept of time delay as an observable in the context of elastic scattering of particles was first heuristically introduced by L. Eisenbud in 1948 [41]. As shown by E. Wigner [42], in the simple case of single channel scattering one can construct a time delay operator by considering the incoming and outgoing wave packets and their relative phase shift $\phi$, and obtain the time delay

$$t = 2\hbar \frac{d}{dE}\phi(E). \tag{1}$$

This expression was extended by F. Smith in order to consider multichannel scattering [43]. For an incoming wavefunction $\psi_{in}$ and an outgoing wavefunction $\psi_{out}$, the scattering matrix $\mathcal{S}$ is such that $\psi_{out} = \mathcal{S}\psi_{in}$. The time delay is then given by

$$\hat{t} \mapsto -i\mathcal{S}^\dagger(E)\frac{d}{dE}\mathcal{S}(E), \tag{2}$$

where the dagger symbol indicates the conjugate transpose of the matrix [6]. The expressions in Eqs. (1) and (2) and related ones historically go under the name of *Eisenbud-Wigner-Smith* (EWS) *time delay* $t_{EWS}$.

The concept of EWS time delay in particle scattering can be extended to describe the photoionization and photoemission processes. The main idea is that photoemission can be considered as a "half-scattering" process, in the sense that there only is the outgoing electron as a scattered wave in the continuum after absorption of a photon, whereas there is no incoming electron wave packet. The fact that the initial state of the particle is a bound state instead of a scattering state is reflected in the expression for the EWS time delay that is obtained from Eq. (2) by writing the $\mathcal{S}$ matrix from perturbation theory applied to the photoemission process of a one-electron system [6]. This leads to the expectation value of the EWS time delay, that is

$$\tau_{EWS} = \hbar\frac{d\phi}{dE} = \hbar\frac{d\measuredangle\left\{\left\langle\psi_f\middle|\hat{H}_{int}\middle|\psi_i\right\rangle\right\}}{dE}, \tag{3}$$

where the missing factor of 2 with respect to Eq. (1) reflects the half-scattering. The letter $\tau$ has been used instead of $t$ to distinguish the half-scattering from the scattering process. In this case, the phase term $\phi$ is the phase (i.e. the argument) of the complex matrix element $M_{fi} = Re^{i\phi}$ describing the photoemission transition: $\phi = \measuredangle\{M_{fi}\} = \measuredangle\{\langle\psi_f|\hat{H}_{int}|\psi_i\rangle\}$.

An important difference between the presented formulas for scattering and half-scattering is that Eqs. (1)-(2) are strictly speaking well defined only for a short-range (i.e. Yukawa-like) potential. On the other hand, the half-scattered electron in photoemission will feel a long-range Coulomb-like potential, because of the interaction with the positive charge left in the system. Therefore, it is required to extend the discussion to long-range potentials, which has been done already for the problem of particle scattering [44, 45]. In order to do so, one needs to introduce the Coulomb potential in the scattering matrix $\mathcal{S}$ and to explicitly express the so-called centrifugal potential $V_{\text{centr.}} \propto \frac{\ell(\ell+1)}{r^2}$, where $\ell$ is the orbital quantum number. This particular dependence on $\ell$ has been recently observed in attosecondstreaking

on WSe$_2$ [46]. The concept of EWS time delay is thus extended to the so-called *Coulomb time delay* $t_{EWS} \longrightarrow t_C$ [6], where one finds

$$t_C = t_{EWS+C} + \Delta t_{\ln} . \tag{4}$$

The term $t_{EWS+C}$ is the actual time delay due to a phase shift because of the scattering process, in strong analogy with the EWS time delay itself, which does not depend on the position **r** of the electron. The additional term $\Delta t_{\ln}$ is a logarithmic correction that takes into account an additional phase shift $\propto \ln(2\mathbf{k}\mathbf{r})$, which describes the Coulomb "drag" from the positive charge left behind felt by the electron with wavevector **k**.

The description of time delays in long-range potentials stays the same when considering the photoemission process. In this case, a measurement of a relative time delay between different $\tau_C$ coincides with the measurement of $\Delta\tau_{EWS+C}$. One should therefore speak of the Coulomb EWS-like time delay, but literature often refers to it as simply EWS time delay $\tau_{EWS}$ (as in this publication). Also, a possible measurement of an absolute time delay implies that the measured time duration is not anymore referred to the actual time zero, but to the time zero corrected by the trivial term $\Delta\tau_{\ln}$.

Another related quantity, $\tau_{EWS}^s$, will be introduced in Section 2. The physical meaning of these time delays will be discussed in Section 4.

## 1.2 Spin polarization in photoemission from spin-degenerate states

A beam of electrons produced in some physical process can have the two possible spin states along a certain direction that are not equally populated. This is called a *spin polarized electron beam*, and the ensemble quantity *spin polarization* along the direction $i = x, y, z$ is defined as

$$P_i = \frac{N_i^\uparrow - N_i^\downarrow}{N_i^\uparrow + N_i^\downarrow} , \tag{5}$$

where $N^\uparrow$ and $N^\downarrow$ is the number of electrons with spin along $i$ being "up" and "down", respectively. As an average quantity, all three spatial components of the spin polarization vector can be determined by performing three sets of measurements.

Certain classes of materials have some electronic states where the electrons have a preferential spin orientation. These states are said to be spin-polarized in the sense of Eq. (5). A typical example are the classical ferromagnets such as Fe, Co and Ni, where the magnetism is due to their 3$d$ electrons [47,48]. Another example are materials where spin-orbit interaction plays a role in the definition of the electronic structure, such as Rashba materials [49–57] and topological insulators [58–61].

In order to probe the spin polarization of photoelectrons emitted from materials with spin-polarized initial states, it is natural to employ the spin- and angle-resolved photoemission spectroscopy (SARPES) technique [38,62]. It is however very important to keep in mind that the measured spin polarization is not necessarily the one of the initial state, but modification of it can occur during the photoemission process. For example, matrix element effects can change or even reverse the direction of $P$ as a function of photon energy or light polarization [63–65]; the diffraction through the surface can be spin-dependent, thus modifying $P$ [66]; the coherent excitation of different spin states can result in spin interference effects [67]. All these possibilities make SARPES results difficult to interpret not only on a quantitative level, because of the requirement of sequential measurements with faster detectors or because of the required sample stability with slower ones, but also on a qualitative level. Nevertheless, the information that can be extracted is highly valuable once such effects are properly considered.

On top of this, there is another subtle effect that becomes an additional correction to $P$, even when spin-degenerate initial states are considered, and which has not often been taken

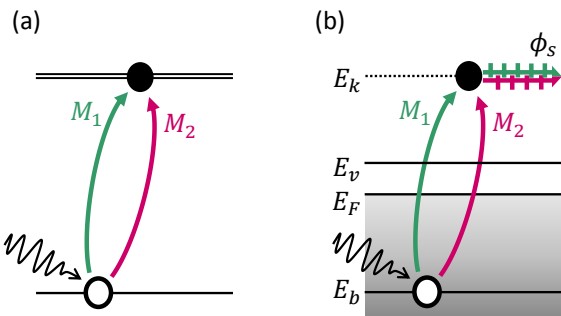

Figure 1: (a) Interference of matrix elements in atomic photoionization for the transition $\ell \to \ell \pm 1$. (b) Two interfering transitions build up the photoemission final state wavefunction, and their phase shift will determine an angular resolved spin polarization.

into account recently, despite being known since almost four decades [27, 33–35, 68–70]: the fact that photoemission can be described as a spin-dependent "half-scattering" process. When studying the process of electron elastic scattering by a central field, in order to account for spin-orbit (SO) coupling one needs to use the relativistic Dirac equation. It is found that SO coupling allows to change the direction of the electron spin upon scattering because of the interaction between the electron magnetic moment and the magnetic field of the scatterer in the electron rest frame, as described by the so-called spin-flip amplitude [33]. As a consequence, it turns out that the scattering of an unpolarized electron beam by a central field leads to a polarized beam depending on the scattering direction [33]. Analogously, a spin polarized photoelectron beam is obtained even in the case when a spin-degenerate initial state is being probed in photoemission [38], as it will be described in more details in the following.

The photoionization of alkali atoms by means of circularly polarized light leads to the emission of electrons that are spin polarized, even when integrated in angle [71]. This is know as the *Fano effect*, and relies on the SO splitting of the atomic levels and on the selection rule $\Delta m_j = +1$ ($\Delta m_j = -1$) for $\varsigma^+$ ($\varsigma^-$) light polarization. By using the spin density matrix formalism, one can find analytical expressions describing direction and modulus of $P$ that depend on the geometry of the experiment. Analogously to the electron scattering, the spin polarization depends on the cross-section for the production of electrons in the two spin channels. As shown by N. Cherepkov [68], it is possible to extend these results to the case of unpolarized light or linearly polarized light by combining the results for $\varsigma^+$ and $\varsigma^-$ light polarization with incoherent or coherent sum, respectively. The interesting result is that, without integrating over all the angles, one does find a spin polarization also in this case, even if there is not a net angular momentum transfer by the incident photon. Also in this case analytical expressions can be found for the spin polarization in atomic photoionization. In particular, the direction of $P$ is perpendicular to the plane defined by electron momentum and light electric field for linearly polarized light (or Poynting vector for unpolarized light), and the modulus of $P$ is found to be $\propto \text{Im}\left[M_1 M_2^*\right]$. This proportionality allows to describe the origin of the spin polarization as the interference term between two transitions to different degenerate final states, described by the matrix elements $M_1$ and $M_2$, as pictured in Fig. 1(a). For example, starting from a level $\ell > 0$, the different degenerate final states are the $\ell + 1$ and $\ell - 1$ levels. This has been experimentally proven for the first time in Xe atoms [72].

The effects seen in atomic photoionization are found also in photoemission from crystals. Angular integrated photoemission by using circularly polarized light, for instance, yields spin polarized electrons when the electronic states involved are influenced by SO coupling, equivalently to the atomic Fano effect. This is, for example, the famous case of the GaAs crystal [73].

Also the use of linearly polarized or unpolarized light can yield spin polarized electrons from spin-degenerate states. As in photoionization, two interfering transitions sensitive to SO coupling are required. This can occur either in a high symmetry setup [74,75] or also when there is a symmetry breaking in the experiment [36,69,76]. In this case in particular, as long as light impinges on the sample surface with off-normal incidence, the photoelectrons are spin polarized both in the case of normal emission and off-normal emission. An intrinsic complication is the fact that with off-normal emission further effects such as diffraction through the surface might play a role; but on the other hand it is necessary to probe off-normal emission electrons if dispersive states are being measured, in order to fulfill the energy-momentum dispersion relationship.

An interesting complementary approach to the study of spin polarization is the use of spin-resolved *inverse photoemission*, which is the time-reversal process of photoemission where spin polarized electrons are sent on a crystal and UV light is emitted upon deexcitation [77,78]. In this case the similarity with a half-scattering process are even clearer, and similar effects to the ones described in this Section have been observed in the unoccupied states [79].

Analogous to the atomic case, one finds that the spin polarization in photoemission from solids is given by the following expression adapted from Refs. [69,76,80]:

$$\boldsymbol{P} = I_{tot}^{-1}(\Omega)f(\Omega)\text{Im}[M_1 \cdot M_2^*]\boldsymbol{n} = P(r, \phi_s)\boldsymbol{n}\,. \tag{6}$$

The term $\Omega$ is a set that contains all the relevant angles describing the actual photoemission geometry. Only the case of linearly polarized light is considered here. As usual, the definition of polarization requires the normalization term $I_{tot}^{-1}$, which will depend on $\Omega$. Also in photoemission from solids, the spin polarization of photoelectrons from spin-degenerate states is given by the interference term $\text{Im}[M_1 \cdot M_2^*]$ between (at least) two photoemission channels, described by the matrix elements $M_1$ and $M_2$ (see Appendix B for a discussion on multiple channels). In particular, the modulus of the polarization $P$ depends on the quantities $r$ and $\phi_s$, i.e. the ratio of the radial part of the two matrix elements $r = R_2/R_1$ and the difference between the two phase terms $\phi_s = \phi_2 - \phi_1$. The phase shift has the subscript $s$ since it is at the origin of the spin polarization, not to be confused with the phase term $\phi$ of Eq. (3).

As for the direction $\boldsymbol{n}$, it will not only be due to the direction of $\boldsymbol{E}$ and $\boldsymbol{k}$, but it will also depend on the symmetry of the particular crystal and state under consideration. This occurs also for photoionization of molecules, where the direction $\boldsymbol{n}$ is influenced by the symmetry of the molecule [81]. Some specific equations have been derived for certain cases in solids [80], but it is more useful here to consider the generic direction $\boldsymbol{n}$, not necessarily known a priori but in principle accessible in an experiment by measuring the three spatial components $P_{x,y,z}$ [39]. The proportionality constant $f$ will also depend on the actual crystal and geometry, and it can be seen as a *geometrical correction term* that depends on $\Omega$ [39] (see Section 2.1).

This effect was theoretically demonstrated by E. Tamura and R. Feder, who showed that a necessary ingredient is the use of a one-step photoemission model [69,74,76]. In fact without the translational symmetry breaking at the crystal surface, the three-step model cannot take into account the interference since both initial and final states are Bloch states. If compared to atomic photoionization, however, the matrix elements are not related to different partial waves in the final state, but to the projection of the linear light polarization electric field vector onto the crystal surface, which will have a parallel and a perpendicular component. The reason is that the two components will allow a transition from or to different spatial parts of the double group symmetry representation of the electronic states, which is necessary to introduce when considering the SO coupling [36,69,82,83]. In a more general sense, the expression in Eq. (6) can be considered as the result for any situation where SO coupling allows the mixing of spin channels and two degenerate transitions are possible in the experiment, either because of different final states or initial states [36]. For instance, in nonmagnetic crystals with inversion

symmetry every state is twofold degenerate [84, 85]. The situation is pictured in Fig. 1(b), where the wavefunction of the measured free photoelectron is build up by the interference of two different transitions and has a phase term that is the phase difference between the two matrix elements. In a multiple scattering picture as in KKR calculations, without further developments of the theory, one can still consider as a simplification that the spin polarization comes out of all the possible interference paths as if dependent on a "net" phase shift $\phi$ (see also Appendix B).

This effect has been experimentally investigated in the past by using circular polarized light [83, 86, 87], unpolarized light [88] as well as linearly polarized light [36, 75, 82, 83], but only on localized states or dispersive states with poor angular and energy resolution. Recently, this effect has been studied in the dispersive *sp* bulk state of a single crystal of Cu(111) [39], without integrating in energy or angle but maintaining the angular and energy resolution that are typical in ARPES. This has allowed to access the phase information via the spin polarization, and thus to make a link to the time delay in the photoemission process. In this manuscript the analytical model that allows this estimate is presented in detail, with all the explicit expressions and with the introduction of two different time delays. For clarity, the values of parameters due to the experimental setup are chosen for the COPHEE end station at the Swiss Light Source [89–91], and the experimental data are the one for Cu(111) [39].

## 2 Phase shift as a common term

In Section 1.1 it was shown how the Eisenbud-Wigner-Smith (EWS) *scattering time delay* depends on the phase term $\phi$ of the matrix element [Eq. 3]. In Section 1.2, on the other hand, it was discussed how the spin polarization in photoemission from spin-degenerate states arises from an interference process between two different channels of the matrix elements. In particular, $P \propto \mathrm{Im}\left[M_1 M_2^*\right](r, \phi_s)$ [Eq. (6)], which depends on the ratio of the radial terms $r = R_2/R_1$ and the phase shift $\phi_s = \phi_2 - \phi_1$.

The two phases $\phi$ and $\phi_s$ are closely related. In fact, given the two interfering channels 1 and 2, one has $M_{fi} = Re^{i\phi} = \left\langle \psi_f \middle| \hat{H}_{int} \middle| \psi_i \right\rangle = \left\langle \psi_f \middle| \hat{H}^1_{int} + \hat{H}^2_{int} \middle| \psi_i \right\rangle = M_1 + M_2 = R_1 e^{i\phi_1} + R_2 e^{i\phi_2}$, and by making the sum of complex numbers in polar form one obtains:

$$\phi = \phi_1 + \arctan\left(\frac{R_2 \sin(\phi_2 - \phi_1)}{R_1 + R_2 \cos(\phi_2 - \phi_1)}\right) = \arctan\left(\frac{r \sin \phi_s}{1 + r \cos \phi_s}\right), \qquad (7)$$

where in the last step it has been chosen $\phi_1 = 0$, since it is necessary to set a reference given that the two phases $\phi_{1,2}$ are not absolutely determined [see discussion in Section 4]. It is useful to note that $r > 0$ since the radial terms are positive, and $\phi$ and $\phi_s$ are defined within $\left[-\frac{\pi}{2}, +\frac{\pi}{2}\right]$.

At this point the possibility of accessing the time information by the measurement of the spin polarization is investigated. In fact, Eq. (7) shows how the measurement of $P$ in photoemission can in principle lead, via $\phi_s$, to an estimate of $\phi$, and therefore of $\tau_{EWS}$ by varying the kinetic energy of the photoelectron $E_k$. In particular:

$$\tau_{EWS} = \hbar \frac{d\phi\left(r(E_k), \phi_s(E_k)\right)}{dE_k} \approx \hbar \frac{d\phi_s}{dE_k} \cdot \frac{r(r + \cos \phi_s)}{1 + 2r \cos \phi_s + r^2} \doteq \tau^s_{EWS} \cdot w(r, \phi_s), \qquad (8)$$

where the approximation consists in considering $\frac{dr}{dE_k} \approx 0$ (see Section 3.1). The EWS *time delay of the interfering channels*

$$\tau^s_{EWS} = \hbar \frac{d\phi_s}{dE_k} \qquad (9)$$

has been introduced, and the rational function $w = w(r, \phi_s)$ has been defined. The physical meaning of $\tau_{EWS}^s$ and $\tau_{EWS}$ will be discussed in Section 4. Before proceeding with the evaluation of the two EWS time delays from the measured spin polarization, it is useful to write the explicit dependence of $P$ in Eq. (6) on $r$ and $\phi_s$, which is done in the following subsection.

## 2.1 Geometrical correction

In Eq. (6), the geometrical correction term $f(\Omega)$ depends on the set $\Omega$ of relevant angles describing the symmetry of the system. As mentioned in Section 1.2, in the case of atomic photoionization with linearly polarized light the direction $\boldsymbol{n}$ of the spin polarization is perpendicular to the *reaction plane* defined by the light electric field vector $\boldsymbol{E}$ and the outgoing electron momentum $\boldsymbol{k}$. The only relevant angle that determines the spin polarization magnitude in atomic photoionization is the angle $\gamma$ between $\boldsymbol{E}$ and $\boldsymbol{k}$. It can be shown [33] that the proportionality coefficient in this case is $f = 4 \sin \gamma \cos \gamma$.

Also as mentioned in Section 1.2, in photoemission from solids the reaction plane can vary, depending on the symmetry of the crystal under consideration [69, 76]. For localized states, one would expect a similar behaviour as for atomic levels, but for dispersive states the situation is different because of an intrinsic symmetry reduction. Whereas it should be in principle possible to determine such direction for specific crystals by symmetry arguments, it is however very difficult in practice. A different approach consists in determining the reaction plane *a posteriori*, by considering it as the one perpendicular to the *measured* spin polarization vector. The geometrical correction term becomes $f = 4 \sin \gamma' \cos \gamma'$, where $\gamma'$ is the angle between the projections of $\boldsymbol{E}$ and $\boldsymbol{k}$ in the reaction plane, and thus depends on the set of relevant angles $\Omega = (\gamma, \theta, \psi, \delta)$ defined in Fig. 2. In the following, the expression for $f$ will be derived as a function of these angles. As an example, the experimental setup of the COPHEE endstation at the Swiss Light Source will be considered, where: the angle between incident light and outgoing photoelectron is fixed (at 45°); $\pi$ polarized light is used; in order to access different points of reciprocal space, a momentum distribution curve (MDC) is measured by rotating the sample normal (dotted line) by the polar angle $\theta$ in the plane $(\boldsymbol{E}, \boldsymbol{k})$.

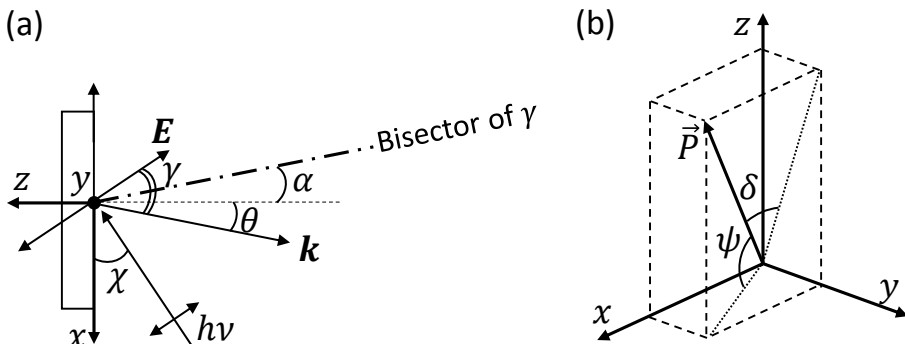

Figure 2: Definition of the relevant symmetry angles: (a) $\gamma, \theta, \alpha, \chi$ and (b) $\psi, \delta$. See the text for details.

In Fig. 2(a) the angles $\gamma$, $\theta$, $\chi = (\gamma - \theta)$ and $\alpha = (\gamma/2 - \theta)$ are shown. Also the sample coordinate system is shown. Since the angle between incident light and outgoing photoelectron is chosen as fixed, also $\gamma$ is fixed ($\gamma = 45°$). The angles $\chi$ and $\theta$ can be used to evaluate the ratios $E_x/E_z = \tan \chi$ and $k_x/k_z = \tan \theta$. In Fig. 2(b) the angles $\psi$ and $\delta$ are shown. They are the elevation angles of the measured $\boldsymbol{P}$ from the $xy$ and $yz$ planes, respectively, and thus are always between 0° and 90°. Accordingly, the three components of the spin polarization vector can be written as: $\left( P_x, P_y, P_z \right) = \left( P \sin \delta, P \sqrt{\cos^2 \delta - \sin^2 \psi}, P \sin \psi \right)$. It is important

to underline that the deviation of $\psi$ and $\delta$ from the atomic case ($\delta = \psi = 0°$) intrinsically depends on the orientation of the crystal planes and the orbital symmetry of the state under consideration, but here they are only considered as the outcome of a measurement.

It is useful to rewrite the correction term $f$ in terms of the parameter $t$ (not time) defined as $t \doteq \tan(\gamma'/2)$, which is commonly known in trigonometry as parametric Weierstrass substitution. This gives:

$$f(\gamma') = 4\sin\gamma'\cos\gamma' = 8t\frac{1-t^2}{(1+t^2)^2}\,. \tag{10}$$

Now it is necessary to evaluate the parameter $t = \tan\left[\gamma'(\gamma,\alpha,\psi,\delta)/2\right]$ by trigonometric construction. One obtains

$$t \doteq \tan\left(\frac{\gamma'}{2}\right) = \tan\left(\frac{\gamma}{2}\right)\left(\frac{\cos\psi}{\cos\delta}\cos^2\alpha + \frac{\cos\delta}{\cos\psi}\sin^2\alpha\right), \tag{11}$$

which can be checked by considering separately the cases where $\psi$ and $\delta$ are zero, first with $\alpha = 0°$ and then varying $\alpha$. Now for a given experiment the coefficient $f$ can be calculated. It has to be pointed out that apart from exceptional cases, the variation of $f$ with $\theta$, which is varied in a measurement, is negligibly small if the $\theta$ range is small (i.e. a few degrees, as it is for an MDC through a band).

In order to explicitly write $P = P(r, \phi_s)$ from Eq. 6, two more ingredients are needed: the interfering term $\mathrm{Im}\left[M_1 M_2^*\right]$ and an expression for $I_{tot}$. The interfering term can be expressed as

$$\mathrm{Im}\left[M_1 M_2^*\right] = \mathrm{Im}\left[R_1 R_2 e^{i(\phi_1-\phi_2)}\right] = R_1 R_2 \sin(\phi_1 - \phi_2) = -R_1 R_2 \sin\phi_s\,. \tag{12}$$

The expression for $I_{tot}$ as a function of matrix elements can be found in Refs. [69,76] as

$$I_{tot} = 2R_1^2 \sin^2\gamma' + 2R_2^2 \cos^2\gamma'\,, \tag{13}$$

which has been modified here with the angle $\gamma'$ instead of $\gamma$, and where the channels 1 and 2 are specified as the two cases of light polarization vector components perpendicular and parallel to the sample surface, respectively. These two components will select different spatial terms of the double group symmetry representation of the state under consideration. Combining Eqs. (10)-(13), finally Eq. (6) can be written as

$$P = \frac{-2\sin\gamma'\cos\gamma' R_1 R_2 \sin\phi_s}{R_1^2 \sin^2\gamma' + R_2^2 \cos^2\gamma'} = \frac{-2\tan\gamma' r \sin\phi_s}{\tan^2\gamma' + r^2} = \frac{-4t(1-t^2)r}{4t^2 + r^2(1-t^2)^2}\sin\phi_s \doteq c(r,t)\sin\phi_s\,, \tag{14}$$

where the parametrization $t = \tan(\gamma'/2)$ and the trigonometric duplication formula $\tan(\gamma') = \frac{2\tan(\gamma'/2)}{1-\tan^2(\gamma'/2)}$ have been used in the second to last step, and the rational function $c = c(r,t)$ has been defined.

For a fixed value of $r$ one has $\max(c) = 1$ for a certain value $t = t'$, and $\min(c) = -1$ for a certain value $t = t''$. The measured value of $t$ depends on the direction $\boldsymbol{n}$, which is described by the angles $\psi$ and $\delta$, dictated by symmetry requirements, and by the angles $\gamma$ and $\alpha$, given by the experimental geometry. At fixed $\gamma$ and $\alpha$, there are several possible combinations of $\psi$ and $\delta$ for which $t = t'$ or $t = t''$. The experimental values of $\psi$ and $\delta$ for the measurements on the $sp$ bulk band of Cu(111) presented in Ref. [39], which are currently the only available precise measurement for the determination of $t$, are one of these combinations for which $t$ and $r$ (the estimate of $r$ is presented in the next Section) give $\max|c| = 1$. Such coincidence might suggest that the symmetry requirements of the crystal are such that the function $|c(r,t)|$ [and thus $P(r,t)$] is maximized.

## 3 Estimate of time delays

In this Section it will be shown how to estimate the interfering EWS time delay $\tau^s_{EWS}$ and the scattering time delay $\tau_{EWS}$ in photoemission from a dispersive state by measuring the spin polarization as a function of binding energy. At the end of the Section, a scheme that summarizes the model can be found.

From Eqs. (8) and (14), it is clear that knowledge of the parameter $r$ is required in order to directly estimate $\tau^s_{EWS}$ and $\tau_{EWS}$, even if it has already been considered to be constant with kinetic energy. In fact, since $P$ depends on both $r$ and $\phi_s$, an independent measurement of $r$ would be required in order to evaluate $\phi_s$.

In principle, this should be accessible by UV photoelectron diffraction (UPD), where one measures the angular distribution of photoelectrons, which also depends on both $r$ and $\phi_s$. This approach however is experimentally very difficult to perform on dispersive states, and for the moment it works sufficiently well only on molecular orbitals [31, 32]. Furthermore, it would be required to combine spin resolution with UPD. Another way to obtain $r$ would be a careful quantitative analysis of linear dichroism, which however is not often feasible because of difficult control of light intensity for different light polarizations. Therefore one needs to estimate the value of $r = R_2/R_1$, and a possibility is $r = E_{\parallel}/E_z$, where it is assumed that the weights of the two interfering terms in the state under investigation are the same. Else, if they are known for example from calculations, they could be taken into account to modify the estimate of $r$.

Once $r$ is estimated, it is possible to proceed to calculate the EWS time delays in the following way. The measurement of $\boldsymbol{P}$ gives information on both $P$ and $\boldsymbol{n}$, which determines $t$. Now $c(r, t)$ is given, and from Eq. (14) one can calculate $\phi_s = \arcsin(P/c)$. In order to vary $E_k$, one could think of varying the incident photon energy $h\nu$, as it has been routinely done for atomic photoionization [34, 35]. For a dispersive band of a solid, however, this leads to the complication of accessing a different point of the Brillouin zone, since it corresponds to varying the probed $k_z$, and it can be an issue when looking at dispersive bands along $k_z$. In general, matrix element effects related to cross-section can lead to strong variations of photoemission intensity with photon energy. Furthermore, there are technical limitations in varying the photon energy in an experiment where high accuracy is required. Changing the photon energy requires the change of many experimental parameters and introduces changes in the photon beam (energy, intensity, polarization, focus, position) as an additional error source. In practice, the set of different photon energies can only be precisely determined if one measures the position of the Fermi level. Nevertheless, there is another way of changing the kinetic energy $E_k$ of the photoelectrons from a dispersive state, that is by looking at different binding energies $E_b$. This can be performed by changing just one experimental parameter with high accuracy. In this manuscript only this approach will be considered. Henceforth a dot will represent the derivative with binding energy: $\dot{\square} \mapsto \frac{d}{dE_b} = -\frac{d}{dE_k}$. Under the assumption that $r$ is a constant with $E_k$, any change of $P(E_b)$ directly corresponds to a change of $\phi_s(E_k)$.

At this point one can thus evaluate $\phi_s$ for various $E_b$, and then compute the interfering EWS time delay as $\tau^s_{EWS} = -\hbar\dot{\phi}_s$. Now, by using Eq. (8), it is possible to compute $w(r, \phi_s(E_b))$ and finally obtain the scattering EWS time delay $\tau_{EWS}$. Noticeably, since $w$ depends on $E_b$, also $\tau_{EWS}$ will. However, given that the value of $P$ and its variation with $E_b$ is expected to be relatively small, such dependence will not be large.

It is insightful to now consider a different approach, where an estimate for a finite lower limit of $\tau^s_{EWS}$ can be found without relying on the knowledge of the value of $r$. Starting from the expression of $P = P(r, \phi_s)$, multiplying by $\hbar$ and applying the chain rule in order to evaluate

the derivative with binding energy gives

$$\hbar\dot{P} = \hbar\frac{dP}{dr}\dot{r} + \hbar\frac{dP}{d\phi_s}\dot{\phi}_s \,, \tag{15}$$

where the derivative with respect to the relevant angles $\Omega$ has been neglected (since the variation of $f(\theta)$ is negligible in a small $\theta$ range, as already discussed). Since $\tau_{EWS}^s = -\hbar\dot{\phi}_s$, this leads to

$$\tau_{EWS}^s = \frac{-\hbar}{dP/d\phi_s}\left(\dot{P} - \dot{r}dP/dr\right) \approx \frac{-\hbar}{dP/d\phi_s}\dot{P} \,, \tag{16}$$

where in the last step the usual approximation $\dot{r} \approx 0$ has been used (see Section 3.1). The explicit expressions of $dP/dr$ and $dP/d\phi_s$ evaluated from Eq. (14) can be found in Appendix A. The result of Eq. (16) will yield to a similar value of $\tau_{EWS}^s$ as with the direct method discussed before, by estimating $r = E_\parallel/E_z$ and evaluating $dP/d\phi_s(r,\phi_s,t)$. However, in order to only evaluate a lower limit for $\tau_{EWS}^s$, one can proceed in the following way. First, the absolute value of both sides of Eq. (16) is taken. The signs of $P$ and $dP/d\phi_s$ determine the sign of $\tau_{EWS}^s$, which in general can be positive or negative, simply meaning a positive or negative delay of the interfering channel 2 with respect to 1. This distinction is however not very interesting, and it is very difficult to make sure that all the possible contributions to signs are properly taken into account, in the formalism as well as in the experiment. It is therefore more useful to look only at absolute values. Then, it is possible to write

$$\left|\tau_{EWS}^s\right| = \frac{\hbar}{|dP/d\phi_s|}\left|\dot{P}\right| \geq \frac{\hbar}{\max|dP/d\phi_s|}\left|\dot{P}\right| = \hbar\left|\dot{P}\right| \,, \tag{17}$$

since $|dP/d\phi_s| \leq \max|dP/d\phi_s| = 1$, where, in particular, the maximum $|dP/d\phi_s| = 1$ occurs for $\phi_s = n\pi$ with $n$ integer and $|c(r,t)| = \max|c| = 1$ (see the discussion of Eq. (14) and the expression of $dP/d\phi_s$ in Appendix A).

This procedure can be extended to the estimate of a finite lower limit for the scattering EWS time delay $|\tau_{EWS}|$ from Eq. (8) in the following way:

$$|\tau_{EWS}| = \left|\tau_{EWS}^s\right||w(r,\phi_s)| = \frac{\hbar}{|m(r,\phi_s,t)|}\left|\dot{P}\right| \geq \frac{\hbar}{\max|m|}\left|\dot{P}\right| \,, \tag{18}$$

where $m \doteq (dP/d\phi_s)/w$. The explicit expression of $m(r,\phi_s,t)$ can be found in Appendix A. In this case, though, this function cannot be maximized for every possible value of $r$ and $t$, since $|m| \to +\infty$ for $(r,t) \to (0,0)$, and therefore it is not possible to directly set a *finite* lower limit for $|\tau_{EWS}|$ from the measurement of $\dot{P}$. However, for given values of $r$ and $t$ which will be different from 0 ($r = 0$ in fact means that there is no interfering transitions, and $t = 0$ is geometrically pathological), it is possible to evaluate $\max|m(\phi_s)|$ and thus find a finite lower limit for $|\tau_{EWS}|$. Also, by estimating $\phi_s(E_b) = \arcsin[P(E_b)/c]$ from Eq. (14) as previously discussed, one can find the actual value of $|m(E_b)|$, and therefore estimate $|\tau_{EWS}(E_b)|$ itself.

It can be useful to consider a way to find also an *upper* limit to the estimates. This is possible by rewriting Eq. (14) as

$$\left|\frac{P}{c(r,t)}\right| = |\sin\phi_s| \leq 1 \,. \tag{19}$$

This inequality can be used to find the range of allowed possible values of $r$ for given $P$ and $t$ from the measurements, without being limited to the assumption $r = E_\parallel/E_z$. Therefore one can use these different values of $r$ in Eqs. (17) and (18) to find the largest possible values of $|\tau_{EWS}^s|$ and $|\tau_{EWS}|$. These values are the upper limits to the estimate of EWS time delays from the measured values of $P$ and $t$ without any assumption on $r$.

Noticeably, from Eq. (8) it can be seen that the function $|w|$ is always limited between 0 and 1 for $\phi_s \in \left[-\frac{\pi}{2}, +\frac{\pi}{2}\right]$, with the curious consequence that $|\tau_{EWS}| < |\tau_{EWS}^s|$. This might be counter-intuitive at first; however, as discussed in Section 4, the suggested physical interpretation of the two EWS time delays is the following: $\tau_{EWS}$ is a purely (half-)scattering time delay, whereas $\tau_{EWS}^s$ accounts for the time delay of the photoemission process. This is because the two interfering partial channels do not correspond to two separate events, but they *together* form the final photoelectron. In addition to this interpretation, it is worth to consider more carefully the allowed range for $\phi_s$, as discussed in the following.

Previously, it has been mentioned that $\phi$ and $\phi_s$ are defined within $\left[-\frac{\pi}{2}, +\frac{\pi}{2}\right]$. Whereas this is true for $\phi$ from Eq. (7), there is no univocal choice for $\phi_s$, which only has to be in a range of $\pi$. If one chooses $[0, \pi]$, the function $|w|$ can have values larger than 1, and therefore it can occur that $|\tau_{EWS}| > |\tau_{EWS}^s|$. Also, the estimate of $\max |m(\phi_s)|$ is not straightforward anymore, since for a given $r$ and $t$ there is now a certain $\phi_s$ for which $|m|$ still diverges. A further complication involves the estimate of $\phi_s$ itself from Eq. (14), which is not anymore univocal either, since also the solution $\phi_s = \pi - \arcsin(P/c)$ becomes possible. It has to be noted, however, that since $M_{fi} = R_1 + R_2 e^{i\phi_s}$ from Eq. (7) the choice of the range $\left[-\frac{\pi}{2}, +\frac{\pi}{2}\right]$ seems more natural, since it allows the two interfering channels to be combined in a way that $\text{Im}\left[M_{fi}\right]$ can be both positive and negative. Whereas this issue deserves further theoretical investigation, it will not be considered anymore in the following for simplicity, and only the range $\left[-\frac{\pi}{2}, +\frac{\pi}{2}\right]$ will be considered.

In Fig. 3 a summary of the model presented so far can be found. In particular it shows that by measuring the spin polarization modulus and direction as a function of binding energy for a certain dispersive state, and with the assumption $\dot{r} \approx 0$, one can estimate the lower and upper limit of the interfering EWS time delay $|\tau_{EWS}^s|$. With the further assumption $r = E_{\parallel}/E_z$, or knowing $r$ from calculations or other experiments such as UV photoelectron diffraction, then one can evaluate $|\tau_{EWS}^s|$ itself. Using Eq. (8) also the scattering EWS time delay $|\tau_{EWS}|$ can be obtained.

## 3.1 Influence of the radial terms on the estimate

In order to find a good estimate of the time delays, in the previous Section it has been discussed how a reasonable choice for the value of $r$ would be $r = E_{\parallel}/E_z$, but other estimates of $r$ are possible. In Fig. 4 the dependence of $|\tau_{EWS}^s|$ and $|\tau_{EWS}|$ on $r$ is shown for a given value of $\dot{P}$ and for different values of $t$ and $\phi_s$, whereas in Fig. 5(a) the values of $\dot{P}$, $t$ and $\phi_s$ are the ones found in the experiment on the $sp$ bulk band of Cu(111) presented in Ref. [39]. Since $\phi_s$ varies with $E_b$, the different plots of Fig. 4 for different $\phi_s$ should be considered when the values of $|\tau_{EWS}^s|$ and $|\tau_{EWS}|$ are evaluated for different $E_b$.

Furthermore, in the previous Section the assumption that the parameter $r = R_2/R_1$ does not vary with binding energy ($\dot{r} \approx 0$) has been made. In the following, this assumption will be discussed. The ratio $r = R_2/R_1$ depends both on the geometry, i.e. on the projection of the $E$ field vector onto the crystal surface, and on the electronic state composition in terms of double group symmetry representation. In order to measure spin-resolved MDCs through a dispersive band at different binding energies, the angle $\theta$ will be different by only a few degrees within the whole band. Thus, since $E_{\parallel}/E_z = \tan \chi = \tan(45° - \theta)$, a small change of $\theta$ will not sensibly affect $r$ for different $E_b$. As for the state composition, in principle one could have a strong variation of matrix elements within the band under special circumstances, for instance along very low symmetry directions or where states are hybridized with neighbouring bands. However, for a well defined band within a small $E_b$ range, it is reasonable to assume that its double group symmetry representation does not sensibly vary when the state is considered along a certain high symmetry direction. Experimentally, a good hint to make the assumption $\dot{r} \approx 0$ is to observe $\dot{I}_{tot} \approx 0$ [see Eq. (13)].

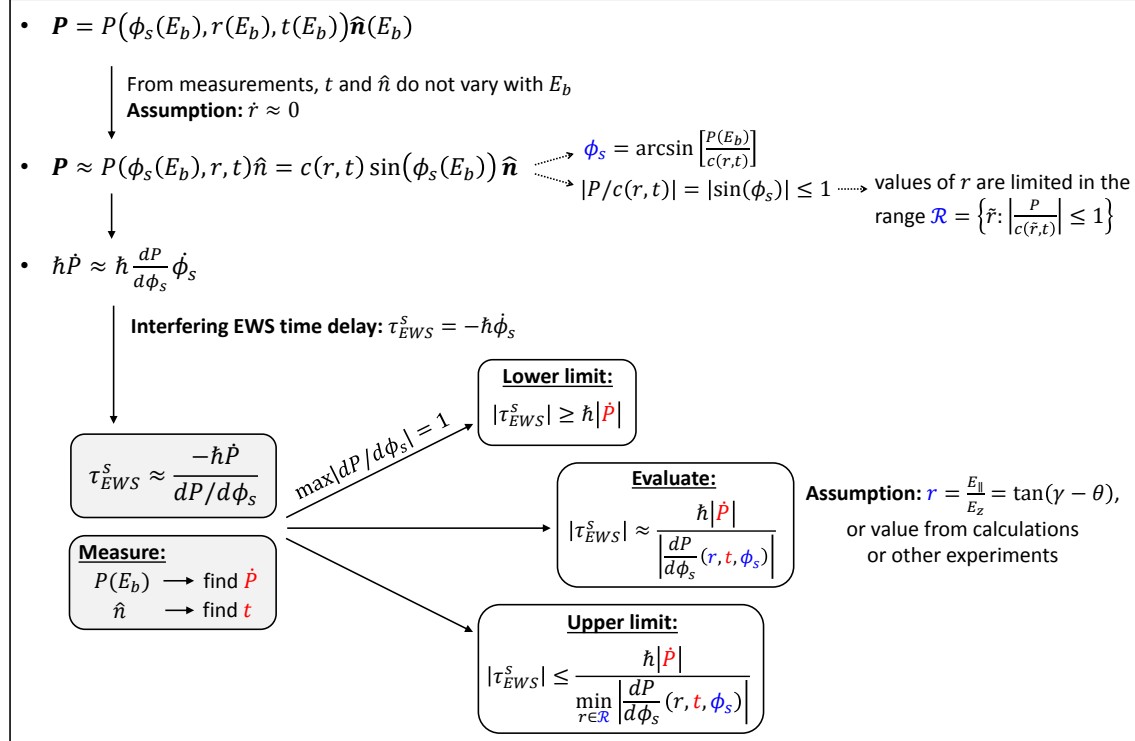

Figure 3: Summary scheme for the estimate of $|\tau^s_{EWS}|$ from the measurement of $\boldsymbol{P}$. Modulus and direction of $\boldsymbol{P}$ are determined by both crystal properties and experimental setup. Measuring $\boldsymbol{P}(E_b)$ and assuming $\dot{r} \approx 0$ one can estimate the lower and upper limits of $|\tau^s_{EWS}|$. With a further assumption for $r$, one can evaluate $|\tau^s_{EWS}|$ itself. See the text and also Ref. [91] for more details.

On the other hand, it is still possible to consider the more general case where $\dot{r} \neq 0$. In such a case, a variation of $P$ with $E_b$ is due to a time delay ($\dot{\phi}_s$), but also to a change of matrix elements ratio within the band ($\dot{r}$), as shown by Eq. (15). Also, Eq. (8) is modified as in the following:

$$\tau_{EWS} = \tau^s_{EWS} \cdot w(r, \phi_s) - \hbar \dot{r} \cdot w'(r, \phi_s) = \frac{-\hbar \dot{P}}{m} + \frac{\hbar \dot{r} \left( dP/dr - w'm \right)}{m}, \qquad (20)$$

where the last step is obtained by inserting the full form of Eq. (16). The explicit expression of the function $w'(r, \phi_s)$ is reported in Appendix A, together with all the other functions that have been introduced. In Fig. 5(b) the effect of $\dot{r}$ on the estimate of time delays is shown, where $|\tau^s_{EWS}|$ and $|\tau_{EWS}|$ are plotted as a function of $\dot{r}$ for given values of $r$, $\dot{P}$, $t$ and $\phi_s$. Noticeably, in this case there are values for which the time delays are zero, which means that the variation of $P$ with $E_b$ is entirely due to variation of the radial part of the matrix elements. The most general situation will correspond to variations in both phase shifts (i.e. time delays) and radial parts. Also, it is important to point out that there exist values of $\dot{r}$ for which $|\tau_{EWS}| > |\tau^s_{EWS}|$.

## 3.2 Spurious effects

In the case of photoemission from solids there might be additional effects other than the interference described in this publication that will modify the spin polarization vector. These spurious effects can be due to diffraction through the surface, or to scattering with defects of the crystal during the transport to the surface. The formalism described until here will be modified in the following way. The spurious effects are modeled by a spin polarization term $\boldsymbol{\eta}$, such

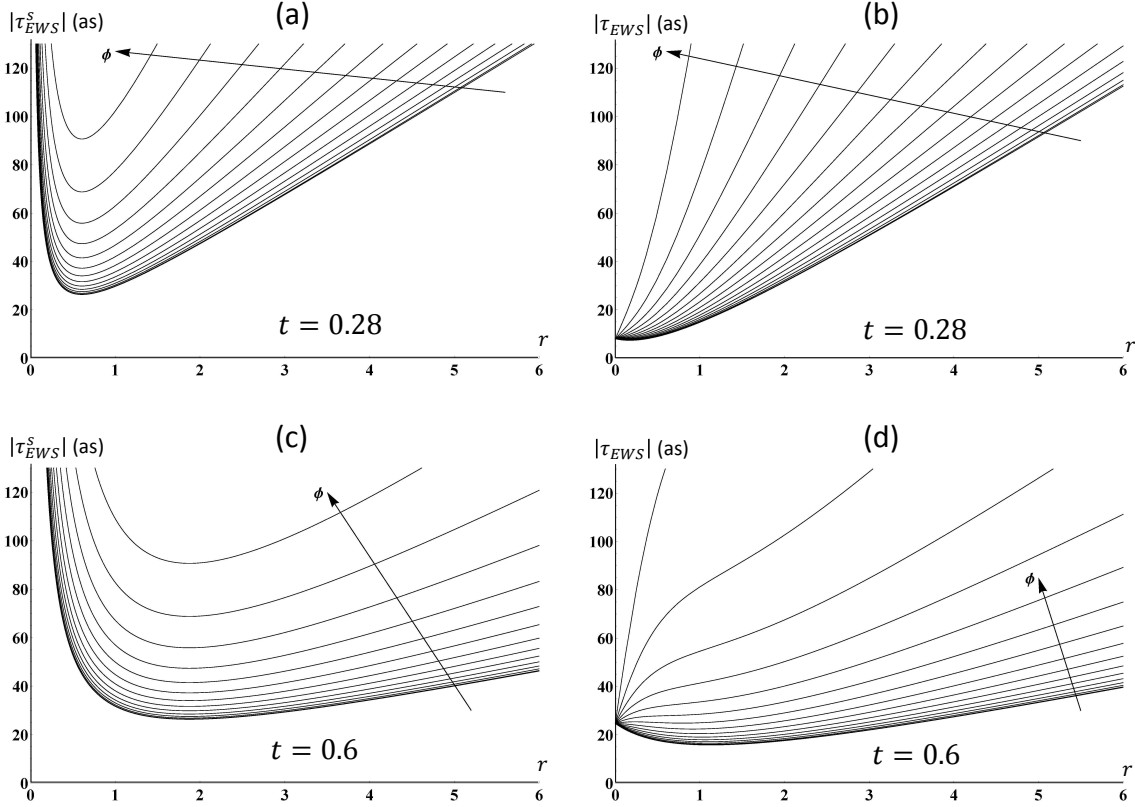

Figure 4: Time delays $|\tau^s_{EWS}|$ [(a), (c)] and $|\tau_{EWS}|$ [(b), (d)] plotted as a function of $r$ for two values of $t$: $t = 0.28$ [(a), (b)] and $t = 0.6$ [(c), (d)]. Every figure shows several plots for different values of $\phi_s$ ranging from 0 to $\pi/2$ (same trend for $\phi_s$ ranging from 0 to $-\pi/2$). The value of $\dot{P}$ is $\dot{P} = 0.04$ eV$^{-1}$.

that the measured spin polarization vector $\boldsymbol{P}_m$ is given by $\boldsymbol{P}_m = \boldsymbol{P} + \boldsymbol{\eta}$, where $\boldsymbol{P} = P\boldsymbol{n}$ is the actual spin polarization given by the interference effects under consideration. The vectorial sum could lead to a rotation of the direction of $\boldsymbol{P}_m$ with respect to $\boldsymbol{n}$, thus making it difficult to determine both the relevant quantity $P$ and the direction $\boldsymbol{n}$ itself, which is used to evaluate the parameter $t$. In fact, one should now write the two components of $\boldsymbol{P}_m$ as $P^n_m = P + \eta^n$ and $P^p_m = \eta^p$, where $n$ and $p$ stand for along and perpendicular to $\boldsymbol{n}$, respectively. Clearly, in general the results presented in the previous Sections are not valid anymore; however, it is possible to make the following two considerations.

- First, one can at least use Eq. (17), where only the derivative of $P$ with binding energy appears, and consider the fact that spurious effects related to scattering will not strongly depend on kinetic energy. Thus it will be possible to proceed with an estimate of the lower limit of $|\tau^s_{EWS}|$.

- Second, diffraction effects will strongly depend on experimental geometry. It is possible to exclude an influence of these effects by measuring the spin polarization from a state of the crystal which is expected not to be dependent on experimental geometry, and showing that there is no variation. This is the case of core levels of the crystal, which will behave as in the case of photoionization of atomic levels [37] so that the spin polarization will not depend on the actual orientation of the crystal with respect to the incoming light. Therefore if this spin polarization does not change when measured with different orientations, it is possible to conclude that the surface does not affect the spin polarization signal with diffraction effects. This situation has been shown for the

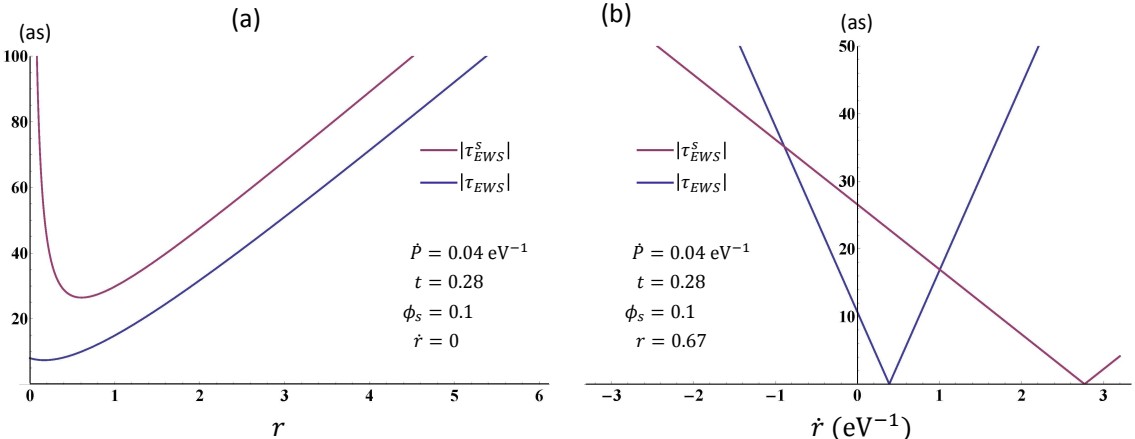

Figure 5: (a) Plot of $|\tau_{EWS}|$ (blue) and $|\tau^s_{EWS}|$ (red) as a function of $r$, for $\dot{P} = 0.04$ eV$^{-1}$, $t = 0.28$ and $\phi_s = 0.1$, where in this case $\dot{r} = 0$. (b) Plot of $|\tau_{EWS}|$ (blue) and $|\tau^s_{EWS}|$ (red) as a function of $\dot{r}$, for $r = 0.67$ and same values as in (a) for the other parameters [from Eqs. (26) and (27)].

experiment on Cu(111) presented in Ref. [39] by looking at the Cu $3p$ core levels.

In the discussion above we have considered all possible influences for the sake of completeness. However, in a carefully designed experiment most of these effects can be eliminated or reasonable assumptions can be made.

## 3.3 The case of a spin polarized initial state

The results presented in the previous Sections concern the spin polarization arising from interfering channels in the matrix elements in the case of a spin-degenerate initial state. The situation is in close analogy with the case of scattering of an unpolarized beam of electrons, which acquires spin polarization because of the interference of the different partial wave components. However, there exist certain classes of materials with an initial state that is already spin-polarized, so that there is a preferential direction in space where the spin of the electrons will point. The question is, how will this spin behave during the photoemission process. There is a number of cases where it has been shown that, indeed, the spin polarization that is measured does not directly reflect the one in the initial state, but it is affected by the particular experimental conditions [65, 92]. It is natural to ask how the interference effect previously presented will affect a spin-polarized state, a question which has not been much investigated in the literature.

A proposal to address the issue of spin polarized initial states is the following. Along the lines of the electron scattering picture, one should consider the case of the scattering of a spin-polarized electron beam. It is reasonable to assume that the spin polarization of photoelectrons will be modified in a similar fashion as in the case of scattering. The behaviour of spin-polarized electrons in elastic scattering is well known in literature [33]. In general, the spin polarization vector of an electron beam rotates and also changes its modulus upon scattering. However, in the particular case where $P = 1$, it will only rotate without modifying its modulus. This is very interesting when considering the analogous case of photoemission as a half-scattering process. It shows that if the photoelectron beam is expected to have a spin polarization $P = 1$, as is often the case [93], the interference effect of the different channels in the matrix elements will not modify the modulus of $\boldsymbol{P}$, but it will only rotate its direction from the expected one. Indeed, in a real experiment, the measured spin polarization very rarely points exactly along the direction that is expected from theory, but it is often canted

by small angles in different directions depending on the actual experimental geometry. Such observation could be explained by taking into account the interference effect described in this publication.

## 4  Discussion and conclusions

The chronoscopy of photoemission is a fundamental topic in modern physics [6]. Time-resolved photoemission experiments show that the time scale of the process is in the attosecond ($10^{-18}$ s) domain, which is the natural time scale of atomic processes. The direct measurement of a finite *relative* time delay between different photoelectron beams suggests the existence of a finite *absolute* time delay of photoemission for each beam, even though this issue has been experimentally addressed only recently and still relies on comparison with theoretical calculations [25, 26].

A different, complementary approach to the chronoscopy of photoemission has been suggested to be through the spin polarization of the photoelectrons [38]. In fact, attosecond time delays in the photoemission process and the spin polarization of photoelectrons are both related to the phase information in the matrix elements[1]. The relationship between spin polarization and time delay in photoemission from solid state targets has been investigated in this publication, with particular focus on dispersive spin-degenerate states. Both quantities can been modeled by considering the photoemission process as an electron half-scattering process, which highlights the central role of the phase shifts. In particular, the relationship between the two phase terms has been shown: $\phi$, corresponding to the full matrix element, and $\phi_s = \phi_2 - \phi_1$, corresponding to the relative shift between two interfering channels. Correspondingly, the two time delays $\tau_{EWS}$ and $\tau_{EWS}^s$ have been introduced. It has been shown how to estimate, under certain assumptions, these two quantities for a dispersive band from the measurement of $\dot{P}$, the variation of spin polarization with binding energy, and with an accurate description of the experimental geometry.

Taking all these points into account we can have a closer look at the experimental data for Cu(111) presented in Ref. [39] and extract values for $\tau_{EWS}$ and $\tau_{EWS}^s$. The various parameters obtained from the measurement of modulus and direction of the spin polarization vector as a function of binding energy $E_b$ for the *sp* bulk band of Cu(111) are: $|\dot{P}| \approx 0.04$ eV$^{-1}$, $r = E_\parallel/E_z = 0.67$, $t = 0.28$ and $\phi_s(E_b)$ varying from 0.1 to 0.2. However it can be checked [see Eq.(23) ] that this variation leads to a negligible variation in the estimate of the time delays, therefore only one can be considered. With all these values, from Eq. (17) one can estimate $\left|\tau_{EWS}^s\right| \geq 26$ as, but also $\left|\tau_{EWS}^s\right| \approx 26$ as from Eq. (16) because of the combined optimum of $r$ and $t$ yielding $c \approx 1$. Similarly, from Eq. (18) one obtains $|\tau_{EWS}| \approx 11$ as. If one does not evaluate $r = E_\parallel/E_z$ but allows different values for $r$, the estimates of the time delays will be different. Then it is possible to set an upper limit as discussed in Section 3.1. From Eq. (19) one finds $r_{max} = 12.1$ by using $\phi_s = 0.1$ and $r_{max} = 6$ by using $\phi_s = 0.2$. Under the assumption $\dot{r} \approx 0$ the largest possible value of $r$ is $r_{max} = 6$, which gives $\left|\tau_{EWS}^s\right| \leq 134$ as and $|\tau_{EWS}| \leq 115$ as. It should be mentioned that the estimate of an upper limit for $\left|\tau_{EWS}^s\right|$ as

---

[1] The interconnection between *time, phase* and *spin* triggers some sort of chicken-or-egg philosophical question. On one side, one could consider the absolute time delay of photoemission as a fundamental property of the process, since it is necessary to have some finite time lag between the initial and the final state, even in the one-step model picture. Then the time delay requires a certain dependence of phases on energy according to Eq. (3), and as a consequence they determine a certain spin polarization according to Eqs.(7) and (6). On the other hand, one could think of the phase term of the matrix element describing the transition to be the fundamental quantity determined by the process, and then, as a consequence, time can be considered as an *emergent* property, at the same level as the spin polarization. This second view, even if less intuitive, has some similarities with other descriptions of the nature of time in different fields [94].

reported here is more precise than the one in Ref. [39]. To summarize, the following estimates of EWS time delays for the *sp* bulk band of Cu(111) have been found: 26 as$\leq \left|\tau_{EWS}^s\right| \leq$ 134 as and 11 as$\leq |\tau_{EWS}| \leq$ 115 as. Furthermore, by assuming $r = E_\parallel/E_z$ one finds $\left|\tau_{EWS}^s\right| \approx$ 26 as and $|\tau_{EWS}| \approx$ 11 as.

For the experimental data for the cuprate superconductor $Bi_2Sr_2CaCu_2O_{8+\delta}$ in Ref. [40] such a detailed analysis is unfortunately not possible, because accurate values for $\left|\dot{P}\right|$ and $t$ cannot be obtained. Therefore it is only possible to give the following estimates for a lower limit: $\left|\tau_{EWS}^s\right| \geq$ 85 as and $|\tau_{EWS}| \geq$ 43 as [91]. These significantly longer time scales compared to Cu raise the question about the influence of correlations on the time delay. However, this requires more detailed measurements and is a topic for further research.

Two points need to be clarified about the applicability of the methodology presented here. First, if in a particular case no interference occurs in the matrix element (for example, $r \to 0$ or $r \to +\infty$), then no spin polarization is produced in the photoemission process. The phase term $\phi_s$ would not be well defined, and therefore $\tau_{EWS}^s$ neither. However, a time delay might still take place, just it would not be accessible by spin-resolved ARPES. Second, whereas this paper has dealt mainly with the case of spin-degenerate initial states, it is possible to extend this approach to the case of spin-polarized states, as outlined in Section 3.3. Interference effects will be concealed though, since they will contribute only to a small degree of polarization when a spin quantization axis is well defined by the physics of the initial state. This remark explains why, whereas SARPES measurement very often confirm on a qualitative level the theoretical predictions made on the physics of the initial state, still a a small rotation of the measured spin polarization away from the expected one is quite common.

Another important comment about the methodology presented is that it allows to extract the time information also from non-time-resolved calculations, as long as spin-resolved one-step photoemission calculations are considered. In fact in this case the photoemission matrix elements are fully described, and therefore the phase information is calculated and processed. This can be very powerful when employed on systems that are experimentally difficult to probe with time-resolved or spin-resolved ARPES, and shows in general that it is possible to improve the understandings of photoemission calculation outputs. This approach could prompt further advances in photoemission theory.

It is important to underline the nature of the interfering transitions responsible for the spin polarization. In the case of atomic photoionization, they correspond to the two final partial waves with orbital quantum number $\ell \to \ell \pm 1$. In solids, on the other hand, they are given by two mixed spatial symmetries of the considered state in the double group symmetry representation, both in the initial and final states [83, 86], that are selected by the in-plane and out-of-plane components of the electric field of the light [88]. In any case, the interfering transitions are different photoemission channels which *do not correspond to different photoelectrons*, as in the case of time-resolved experiments, but *they together build up the photoelectron wavefunction*. As analogy, one could think of the well-known double slit experiment: also in this case the interference does not occur for two different particles, but each single particle has a behaviour that is result of the interference of the different possible paths. A similar interference mechanism has been proposed for a double slit experiment in the time domain instead of the space domain, for below threshold photoemission experiments using phase-stabilized few-cycle laser pulses [95]. Also in this case the interference is described between different channels of the same photoelectron passing into different time slits, and not between two different photoelectrons.

An interesting point of view on time delays in photoemission is given by the so-called time-dependent configuration-interaction with single excitations (TDCIS) calculations [96]. It has been shown that the coherence of the hole configurations in atomic attosecond photoionization is strongly affected by pulse duration and energy, as well as by the interaction of the ion with

the outgoing electron [97]. This is because the so-called interchannel coupling mechanism [98], i.e. the interference of different ionization channels mediated by the Coulomb interaction with the electron, results in a enhanced entanglement between photoelectron and ionic system, which last for a time delay in the attosecond domain [97]. An equivalent coupling mechanism should occur in dispersive states of a solid, where in addition intrinsic plasmonic satellites [99] might play a role.

At this point it is necessary to discuss the physical meaning of time delay in photoemission, and in particular the two quantities $\tau_{EWS}^s$ and $\tau_{EWS}$. In the three-step model of photoemission, it is easy to identify at least one step where a time delay takes place, that is in the second one. However, this travel time of the electron during the transport to the surface [99, 100] should not be considered in the model for the EWS time delays presented here. In fact, the additional logarithmic correction term corresponding to $\Delta t_{\ln}$ in Eq. (4) strongly depends on the distance traveled by the electron, but it does not give any contribution to the interfering channels since it takes place only once the photoelectron is formed. This is reflected in the fact that the measured $\dot{P}$ does not change for different kinetic energies of the electrons, at least within the experimental capabilities, as observed for the *sp* bulk band of Cu(111) in Ref. [39]. In literature, the interpretation of time delays in photoemission is often made in terms of particle trajectories, since the description of EWS time delays can be made in close analogy with classical mechanics [46, 101]. Whereas this approach elucidates the meaning of the EWS formalism, ultimately one should consider that the three-step model and classical trajectories are just a simplification, and as such they should be extended to a fully quantum description.

In this sense, the one-step model is a better candidate, even though it has only been developed for the description of the energetics of the photoemission process and not its dynamics. In fact, it is difficult to tell which process among photon absorption, electron virtual transition and actual photoelectron emission might occur in a finite time. Indeed the influence of the time evolution of the $E$ field on the phase shifts is under debate [102–104] and there might exist a time-threshold for light absorption. A finite decoherence time required by the wave-function to be formed in the final state might also be an issue to consider [105]. Lastly, once the final state wavefunction is formed above the vacuum level the electron could spend a finite "sticking" time before reaching the free-particle state. In other words, the final state wavefunction will evolve in time such that the density of probability will move from the absorber site towards the outside of the crystal (in analogy with the tunneling process, where the particle wavefunction is already present on both sides of a potential barrier). The last part seems to be the one that better matches the half-scattering picture, but this separation is only artificial, since the process takes place as a whole.

Mathematically, the two EWS time delays $\tau_{EWS}^s$ and $\tau_{EWS}$ correspond to the time delay between the interfering channels and the time delay of the scattering process in the sense of EWS, respectively. However, as already mentioned, the two interfering partial channels do not correspond to two separate events, but they together form the final photoelectron, and thus the interfering time delay should be associated to the time scale of the whole process. In Refs. [106,107] it is discussed how an EWS time delay in photoemission only takes into account the pure scattering delay when the whole phase term is considered. Thus $\tau_{EWS}$ corresponds to $t_{EWS+C}$ of the scattering model [see Eq. (4)]. On the other hand, if an interference phase term is considered, then the time delay can be seen as a formation/release time and not only as a scattering delay. In Refs. [106, 107] the interference is considered between two photon transitions. *It is speculated here that the same idea holds for the interfering partial channels of the matrix elements*. Therefore, $\tau_{EWS}$ accounts for a time delay that is purely due to the (half-)scattering part, whereas $\tau_{EWS}^s$ is the time delay of the actual photoemission process as a whole. This argument can explain why, as mentioned in Section 3, $|\tau_{EWS}^s|$ is actually always larger than $|\tau_{EWS}|$ for $\phi_s$ within the chosen range $\left[-\frac{\pi}{2}, +\frac{\pi}{2}\right]$. On the other hand, as mentioned

in Section 3.1, there exist the possibility that $|\tau_{EWS}| > |\tau^s_{EWS}|$ for certain values of $\dot{r} \neq 0$, therefore the relationship between the two time delays certainly deserves further theoretical investigations. In particular, it is important to notice that in Eq.(7) the choice $\phi_1 = 0$ has been made. Therefore both $\phi$ and $\phi_s$ are defined with respect to the same reference, such that $\phi_s = \phi_2$. Thus $\tau^s_{EWS}$ refers to the time delay between the partial channels that together build up the final photoelectron, and can be seen as a description of the time scale of the photoemission process, i.e. the time it takes for the formation of the photoelectron from the interference of different transitions.

The possible different signs of all the phases lead to time delays that can be positive or negative. In the electron scattering model, the time delay can indeed be negative, meaning that the process is such that the actually scattered electron leaves the sticking region earlier than an electron that would not feel the scatterer potential. This would be correct for the quantity $\tau_{EWS}$, whereas $\tau^s_{EWS}$ should always be positive given the interpretation presented here. In fact the meaning of $\tau^s_{EWS} < 0$ is just that $\phi_2 < \phi_1$, which does not have a direct physical significance. Because of this, and because of the difficulties of carefully determining all the possible sources of a positive or negative sign of the phases in the model and in the experiment, only the absolute value of the EWS time delays has been considered.

A clear limit of the indirect access to time delay presented in this publication is the necessity of having *quantitative* information about the spin polarization for possibly *all the three spatial components*. Given the extremely low efficiency of spin detectors, the required experiments are highly time-consuming, and therefore a systematic approach is rather difficult.

Finally, a possible future development is the following. By combining time-resolved techniques with the measurement of spin polarization, one could cross-compare the different estimates of time delays and have a reliable reference for time-zero. An *attosecond- and spin-resolved photoemission experiment* could allow to time the formation of the spin polarization during the photoemission process, tackling the entanglement of the photoelectron with the photohole left in the system, and thus shedding light on the meaning of time delays in quantum mechanics on a very fundamental level.

## Acknowledgements

The authors would like to thank the following people for useful comments and discussions: M. Donath, U. Heinzmann, R. Kienberger, K. Kuroda, J. Minár, W. Pfeiffer.

**Funding information**   This work was supported by the Swiss National Science Foundation Project No. PP00P2_144742 and PP00P2_170591.

## A   Expressions for the estimate of time delays

In Sections 2-3 several functions have been defined but not calculated, since they are lengthy and not very insightful in their explicit form. For completeness, they are reported in the following. By taking the derivative of Eq. (7) with respect to kinetic energy one obtains eq. (20), where the two functions $w(r, \phi_s)$ and $w'(r, \phi_s)$ are given by:

$$w(r, \phi_s) \doteq \frac{r(r + \cos \phi_s)}{1 + 2r \cos \phi_s + r^2} , \tag{21}$$

$$w'(r, \phi_s) \doteq \frac{\sin \phi_s}{1 + 2r \cos \phi_s + r^2} . \tag{22}$$

In Eq. (14) the function $c(r, t)$ has been defined, such that $P = c(r, t) \sin \phi_s$, where $t \doteq \tan(\gamma'/2)$. In the expressions for the estimates of the EWS time delays, it is necessary to evaluate the derivatives $dP/d\phi_s$ and $dP/dr$. They are given by:

$$\frac{dP}{d\phi_s} = c(r, t) \frac{d \sin \phi_s}{d\phi_s} = \frac{-4t(1-t^2)r}{4t^2 + r^2(1-t^2)^2} \cos \phi_s, \tag{23}$$

$$\frac{dP}{dr} = \frac{dc(r, t)}{dr} \sin \phi_s = \frac{-4t(1-t^2)\left[4t^2 - r^2(1-t^2)^2\right]}{\left[4t^2 + r^2(1-t^2)^2\right]^2} \sin \phi_s. \tag{24}$$

The function $m \doteq (dP/d\phi_s)/w$ is introduced in Eq. (18) for the estimate of $|\tau_{EWS}|$. Its full expression is given by:

$$m(r, \phi_s, t) = \frac{-4t \cos \phi_s (1-t^2)(1 + 2r \cos \phi_s + r^2)}{(r + \cos\phi_s)\left[4t^2 + r^2(1-t^2)^2\right]}. \tag{25}$$

Finally, Eqs. (16) and (20) give the expressions for $\tau^s_{EWS}$ and $\tau_{EWS}$. By combining all the previous equations, one obtains the following explicit forms:

$$\tau^s_{EWS} = \hbar \dot{P} \frac{4t^2 + r^2(1-t^2)^2}{4rt(1-t^2)} \sec \phi_s + \hbar \dot{r} \frac{4t^2 - r^2(1-t^2)^2}{r\left[4t^2 + r^2(1-t^2)^2\right]} \tan \phi_s, \tag{26}$$

$$\tau_{EWS} = \frac{\left[4t^2 + r^2(1-t^2)^2\right]^2(r + \cos\phi_s)\sec\phi_s}{4t(1-t^2)(1 + 2r\cos\phi_s + r^2)} \left\{ \hbar\dot{P} + \hbar\dot{r} \frac{4rt(1-t^2)\left[4t^2 - r^2(1-t^2)^2 - 2r(1-t^2)^2\cos\phi_s\right]\sin\phi_s}{(r + \cos\phi_s)\left[4t^2 + r^2(1-t^2)^2\right]^2} \right\}, \tag{27}$$

where the dependence on $\dot{P}$ and $\dot{r}$ has been highlighted. As an example, for $t = 0.28$, $r = 0.67$ and $\phi_s = 0.1$ [values obtained for the $sp$ bulk band of Cu(111) from Ref. [39]] these last two equations yield $\tau^s_{EWS} \approx \hbar\dot{P} - 0.01\hbar\dot{r}$ and $\tau_{EWS} \approx 0.4\tau^s_{EWS}$, as plotted in Fig. 5(b) for $\dot{P} = 0.04$ eV$^{-1}$.

## B  Multiple channel interference

A more complex scenario can be considered where more than two channels are available and interfere in the photoemission process. Only two channels have been considered for simplicity, and the results can still be applied to a more general case with the simplification of considering two virtual channels that will mimic the actual more complex process. It is in principle possible to analytically expand the model presented to a multiple channel scenario, however the expressions would become very heavy. It would require a modification of Eqs. (7) and (6) for the description of the phases and the spin polarization, respectively. Also, a generalization of Eq. (3) for the definition of the time delay is required [6]:

$$\tau_{EWS} \approx \frac{\sum_q \hbar \frac{d\angle\{M^q_{fi}\}}{dE} \left|M^q_{fi}\right|^2}{\sum_q \left|M^q_{fi}\right|^2}, \tag{28}$$

where the sums are carried out for all the quantum numbers $q$ of the remaining electrons system in the final state. The approximation consists in neglecting the energy derivative of the radial part $R$ of the matrix elements (similarly to what it is discussed in Section 3).

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
