# Peer review of "Determination of the time scale of photoemission from the measurement of spin polarization"

_SciPost Physics, doi:SciPost Phys. 5, 058 (2018)_

## Round 1 · Referee Report · Anonymous (Referee 1) · 2018-8-1

Strengths

adds important information to the previously published experimental data [39] (up to page 11 and App. A)

Weaknesses

1 - contains too many unnecessary considerations which overshadow central result and its application to actual experimental data
2 - after the derivation of the time delays I would have expected suggestions how to measure or estimate the unknowns (in part done in [39])

Report

I have a split opinion on the manuscript "Determination of the time scale of photoemission from the measurement of spin polarization" by Fanciulli and Dil. On the one hand, I am thankful for more elaborate follow-up papers of PRLs and other letter-style papers, on the other hand, there is not enough additional information in the manuscript (apart from an exceedingly long and in part confusing text) to warrant publication.

It starts with the concept of "interference". Since the electrons are distinguishable (spin) there is no two-path interference in this particular setup. Instead, as explained in Refs. [69,76,80], different initial and final states which may also include the direction of the outgoing electron will lead to different matrix elements and the observed polarization as shown in [72] for photoionization from Xe p-states. For this particular case the relative delay of the two contributing partial waves (s- and d-waves) can be evaluated and even classically estimated individually [6]. At this point (bottom of page 6: "The effects seen in photoionization are found also in photoemission from crystals"), a discussion on the matrix elements as in [69,76,80] and their influence on the result of Eq. (7) would have helped which is done implicitly in Sec. 2.1 by geometrical considerations influencing the polarization. Now, Eq. (14) could be inserted in Eq. 7 (side remark: the function $c(r,t)$ has two maxima/minima with $c=\pm 1$) and evaluated for the data at hand (appendix A and top of page 19). If the authors had stopped here this would have been a nice paper with a very general result applied to a specific experimental situation, add a short discussion about error estimates and that's it.

Instead, the second half of the manuscript contains a very general, lengthy discussion on possible time delays under various assumptions without clear indication why the reader should bother. And why should one vary the binding energy instead of the photon energy to change the kinetic energy of the outgoing electron? The argument of the Brillouin zone is not valid as the crystal structure should enter both the initial and final states of the electron (that is one of the reasons for the observed angular dependence of polarization) and why should a procedure become simpler by changing both initial and final states instead of the final state alone? Why should this have a smaller effect on the matrix elements than the "conventional" change of photon energy? This is not clear to me and adds unnecessary confusion in the manuscript as, of course, the photon energy has been changed in experiment (at least that is how I interpret Fig. 2a of [39]).

Finally, EWS(s) and EWS time delays are evaluated which we have learned are complicated functions of various parameters. Fig. 4 shows the delays as functions of r (are these results of the evaluation of Eqs. (26) and (27) from appendix A? Or is it silently assumed that $\dot r=0$? This is not indicated in the text. Where does the energy dependence of the polarization come from? Why the choice of other parameters? We find a reference to the experiment [39] only for the parameter choice in Fig. 5.). We see in Figs. 4 and 5 that any value from 0 to infinity appears to be possible (we have long dropped any signs) with uncertainties ("spurious effects") in all directions. It would probably have helped to analyze matrix elements as in [69,76,80] to assess the influences of different effects.

All in all, half of the paper would have been nice to read, maybe including an extended discussion on the derivation of the equations from [69,76,80]. Application to the data from [39], estimate for the error bars and then stop. All the rest of the paper is questionable, confusing, (in my view) unnecessary.

Requested changes

I recommend to revise the paper by making it more concise and strongly reducing its second half. Instead, a discussion of ways to experimentally determine the unknowns (e.g., discuss the experimental setup of [39]) and estimate remaining uncertainties would be interesting.

---

## Round 2 · Referee Report · Anonymous (Referee 1) · 2018-10-12

Strengths

1 - very interesting results
2 - thorough discussion of calculation procedure
3 - nice presentation of experimental geometry

Report

I understand the authors wish to keep large parts of their work and support this decision. Then, however, I recommend to change the abstract which makes the impression this was some kind of supplemental material to their PRL while the paper is more a review of the method. Changing the second part of the abstract to "... The analytical model for estimating the time delay by measuring the spin polarization is reviewed in this manuscript. In particular, ... of photoemission. The method is then applied to recent experimental data giving upper and lower bounds for the EWS time delay in ..."

But I have a more serious problem with the abstract and also the discussion of the physics in the main text: I still don't see the interference the authors are referring to. Starting from a coherent superposition of spin-up and spin-down states reflecting the initial polarization I will have different oscillator strengths for the transition to the outgoing LEED state (spin-dependent final density of states -> photon-energy dependent polarization) but though both parts of the final wavefunction will end up at the same energy they are still "separated" by their spin. Only if spin-flipping excitations would be available I would get a two-way interference (e.g. up -> up interfering with down -> up). This, however, will not happen for simple application of the dipole operator in Fermi's golden rule. Instead, I could expect to observe the signatures of the coherent excitation adding phase terms to the whole process. The same is true for Fig. 1a where any observable oscillations are signatures of coherence but not interference.

At least this is my understanding of these two terms and the authors should either add a discussion which two pathways are actually interfering or change their wording to reflect the coherence of the excitation process (if applicable). The result does not change but the origin of the effect should be properly discussed.

Requested changes

1 - consider my suggestions in the report part and change manuscript accordingly (if necessary)

---

## Round 2 · Author Response

We thank the referee for reading the manuscript and commenting on it. However, it appears to us that the referee despite, or because of, her/his large background knowledge has misinterpreted the main message of our manuscript and the complications involved in the measurement and analysis.

The referee suggests that we could have stopped after the first part of the manuscript (up to section 3) and then apply this method to our previous results. This suggestion appears to be based on the assumption that the ratio of the matrix elements (r) can be analytically calculated. However, such an analytical approach is only possible for atomic (and maybe molecular) orbits, but not for bands and certainly not away from high symmetry points of the Brillouin zone. In section 3, which the referee regards as questionable, confusing, and unnecessary, we show how in this general case r can be measured (or estimated) and one can thus still obtain a value (or estimate) of the time delay. Here we also include any assumption that might be made and any spurious effect that might have an influence. We consider this inclusion to be important for such a manuscript which goes beyond a letter style paper, especially as indicated in the first paragraph by the referee.

The reason that we change the binding energy instead of the photon energy, is besides the points mentioned in the manuscript, mainly technical. Any SARPES set-up is by definition capable of varying the measured binding with high accuracy, but varying the photon energy is often less trivial. This is clear for laser or discharge lamp based set-ups, but also at a synchrotron the set photon energy can only be accurately determined if one measures the position of the Fermi level. Furthermore, changing the photon energy requires the change of many parameters whereas changing the binding energy requires to change just one experimental parameter, which can be detrimental for experiments where high accuracy is needed. In general, changing the photon energy introduces changes in the photon beam (energy, intensity, polarization, focus, position) as an additional error source. (These points are now included in the manuscript)

As for some other questions the referee raises: - yes it is at some points assumed that \dot{r}=0 as clearly indicated in the text. -The energy dependency of the polarization is the whole topic of the manuscript. It is due to the change in phase, or due to the time delay, depending on how one prefers to look at it (footnote 1).

As a last remark. We don’t agree with the referee that the “electrons are distinguishable (spin) [and therefore] there is no two-path interference in this particular setup.” We are here not talking about electrons that interfere as this would require the simultaneous emission of two electrons in the same energy momentum window which would cause extreme space charge effects. We are talking about the interference of different pathways similar to the single electron double slit experiment. This interference causes the measured spin polarization, which is one of the main points of this and other manuscripts. Furthermore, it has been shown before (ref 65 and 67 for example) that transition channels with different spin expectation values can interfere.

We have not made any further changes to the manuscript based on the referee report. One point that worries us more than any scientific differences of view between the referee and ourselves, which can be resolved, is that the editor required only minor corrections although the referee report was suggesting major changes. And especially that such a decision was based on only a single referee report. We understand the difficulties in setting up a new journal, and we fully support the initiative, but we expected a scientifically more professional approach.

---

## Round 2 · List of Changes

A paragraph concerning the technical reason for the choice of keeping a fixed photon energy in the experiment has been added in Section 3.

---

## Round 3 · Author Response

We thank the referee for the suggestions and for appreciating our work.

We agree with the recommendation about the abstract and have changed it accordingly. This manuscript is not only a review, however, since at the time of the PRL the method was not fully developed as it currently is (e.g., the distinction between scattering EWS and interfering EWS time delays was not made). We also agree that the discussion about interference needs clarification since it is a crucial point, and therefore we have made several changes in Section 1.2, but have kept the wording, given what it is argued in the following where the referee's concerns are addressed.

It is true that the term "interference" might be misleading, because for example the word interference was not used in E. Tamura et al., PRL 59, 934 (1987), where this effect was first properly described for solid targets (as well as in some of the previous similar works on atoms). Nevertheless, in several other publications the spin polarization is said to be due to: - "interference of the wave functions of continuum states reached by the photoionization process" [U. Heinzmann, G. Schonhense, J. Kessler, PRL 42, 24 (1979)], in the case of atomic targets, e.g. the s and d shells reached by an electron from a p shell. - "consequence of a quantum mechanical interference between different photoelectron partial waves", and also "interference of these continuum waves [i.e. transitions from one initial state into two different but energetically degenerated continuum states], which is nonzero if a phase-shift difference exists between the two waves" [N. Irmer et al., PRB 45,7 (1992)]. Similar wording are found in other old references, as well as in the more recent one: - "a quantum mechanical interference of two outgoing partial waves describing the photoelectron emission" [U. Heinzmann and H. Dil, J. Phys.: Condens. Matter 24, 173001 (2012)]. The point is that the term Im[M1M2*] proportional to sin(phi2-phi1) is interpreted as an interference between M1 and M2, and we decided to keep the wording extensively present in the literature. What is more important, is that now we have better addressed the central role of spin-orbit coupling. In fact, as correctly stated by the referee, application of the dipole operator in Fermi's golden rule does not allow spin-flip. Nevertheless, spin mixing in the partial channels becomes possible when spin-orbit coupling is accounted for by relativistic equations (for explicit description see J. Kessler, Polarized electrons 2nd edition, 1985, page 32 for electron scattering and section 5.2.3 for atomic photoionization). This is now stated in the present manuscript, with more stress on the role of SO.

We are confident that we have clarified the referee's concerns and that the manuscript is now ready for publication.

With best regards, Mauro Fanciulli and Hugo Dil

---

## Round 3 · List of Changes

Several sentences of Section 1.2 have been modified, in order to better describe the interference process. In particular: - the possibility of spin-flip due to spin-orbit coupling is mentioned when describing the electron scattering process - spin-orbit coupling is highlighted as a key ingredient in order to have the spin polarization in the photoelectron beam from a spin-degenerate state

---

## Round 4 · Author Response

Dear Editor,
According to your last request, Figure 3 has been simplified and better explained in the caption and the corresponding text. We thank you for pointing this out, since we believe that now the readability of the chapter has improved.
With kind regards,
M. Fanciulli and J. H. Dil

---

## Round 4 · List of Changes

- The scheme of the model for the estimate of time delay in Figure 3 is simpler and clearer than previously.
- Corresponding caption and main text about Figure 3 have been improved.
- The sentence about phases in the conclusions (page 21) has been modified in order to better convey its message.

You are currently on this page

Resubmission 1806.05895v4 on 26 November 2018
Resubmission 1806.05895v3 on 24 October 2018

---

## Editorial Decision

published